# EEG-Based Emotion Classification in Financial Trading Using Deep Learning: Effects of Risk Control Measures

**DOI:** 10.3390/s23073474

**Published:** 2023-03-26

**Authors:** Bhaskar Tripathi, Rakesh Kumar Sharma

**Affiliations:** School of Humanities and Social Sciences, Thapar Institute of Engineering and Technology, Patiala 147004, India

**Keywords:** behavioral finance, emotion classification, deep learning, electroencephalography (EEG), neuro-finance, decision-making

## Abstract

Day traders in the financial markets are under constant pressure to make rapid decisions and limit capital losses in response to fluctuating market prices. As such, their emotional state can greatly influence their decision-making, leading to suboptimal outcomes in volatile market conditions. Despite the use of risk control measures such as stop loss and limit orders, it is unclear if these strategies have a substantial impact on the emotional state of traders. In this paper, we aim to determine if the use of limit orders and stop loss has a significant impact on the emotional state of traders compared to when these risk control measures are not applied. The paper provides a technical framework for valence-arousal classification in financial trading using EEG data and deep learning algorithms. We conducted two experiments: the first experiment employed predetermined stop loss and limit orders to lock in profit and risk objectives, while the second experiment did not employ limit orders or stop losses. We also proposed a novel hybrid neural architecture that integrates a Conditional Random Field with a CNN-BiLSTM model and employs Bayesian Optimization to systematically determine the optimal hyperparameters. The best model in the framework obtained classification accuracies of 85.65% and 85.05% in the two experiments, outperforming previous studies. Results indicate that the emotions associated with Low Valence and High Arousal, such as fear and worry, were more prevalent in the second experiment. The emotions associated with High Valence and High Arousal, such as hope, were more prevalent in the first experiment employing limit orders and stop loss. In contrast, High Valence and Low Arousal (calmness) emotions were most prominent in the control group which did not engage in trading activities. Our results demonstrate the efficacy of our proposed framework for emotion classification in financial trading and aid in the risk-related decision-making abilities of day traders. Further, we present the limitations of the current work and directions for future research.

## 1. Introduction

Emotions are physiological states associated with the neurological system that influence feelings and rational behavior [1]. Emotions and their management are seen as essential factors for efficient and intelligent decision-making [2]. Affective computing is an area of artificial intelligence that focuses on human-computer interactions, such as recognizing human behavior and emotional states [3].

In recent years, automatic emotion detection has been applied in various areas such as emotion recognition from movies [4], audio [5], text [6], and facial expressions [7]. With the development of low-cost wearable technology, non-invasive electroencephalography (EEG)-based techniques for automatic emotion identification have achieved widespread popularity and acceptance [8]. With their high temporal precision, EEG waves directly represent the brain’s neural activity. The information these signals provide is more dependable than that provided by facial expressions or text since they cannot be falsified or replicated to feign an emotional state [9]. High-speed computing has allowed machine learning techniques to be applied with EEG data to detect emotions even more accurately [10]. EEG has been widely used in areas such as medical research to assess brain function and neurological conditions. In recent years, machine learning (ML) models have been applied to EEG signals with explainable AI to develop prediction models for diseases such as stroke and to evaluate mental workload. Examples of such applications include based stroke prediction [11,12], assessment of task-induced neurological outcomes after stroke [13], detection of driving-induced neurological biomarkers [14], and prediction of sleep stages based on EEG biomarkers [15]. In emotion recognition, each emotion combines valence (a spectrum of negative to positive emotions) and arousal (the intensity) associated with the nervous system [16].

In financial markets, day traders are motivated to profit from the market’s price movement in the shortest time feasible while avoiding capital loss. During high-pressure trading sessions, they seek to adjust to changing market conditions and respond to fluctuating market conditions, such as volatility, trend reversals, market sentiment, and events, such as profits and losses. These responses consistently elicit feelings including fear, greed, hope, calmness, and regret [17,18]. Depending on the trade performance, the emotional intensity of traders fluctuates at various periods and may influence their rational decision-making [19]. Day traders operate in a volatile market environment and make use of market price movements to enter or exit their positions. However, they are constantly pressured to make split-second judgments. An erroneous entry or exit decision can lead to significant capital loss [20]. Understanding and classifying their emotions can help them comprehend their emotional state and reactions to various events and make rational decisions to minimize future losses. The psychological and emotional effects of simulated trading can be comparable to actual trading, as most traders attach emotional importance to money-related decisions [17]. It has been observed in past literature that some professional traders tend to hold their positions for too long and sell their winning investments too soon. In addition, many traders apply limit orders to obtain better prices for their assets [21] and use stop-loss strategies to limit excessive losses [22].

Being confronted with regular market volatility, risks, and significant pressure to generate profits while avoiding loss, traders require a systematized framework with ambient intelligence to comprehend their emotional state when making crucial financial judgments to optimize their rational decision-making capacity under market risks [23]. In recent years, many studies have applied deep learning techniques to analyze EEG signals and use them for emotion classification [8,24,25,26,27]. However, the literature on the application of EEG-based analysis in the context of the stock market and financial trading is relatively limited [10,17,28,29]. There is a need for research on the identification and classification of the emotions of day traders in order to gain a deeper understanding of their emotional states and reactions to the results of their trading decisions. Such understanding has the potential to aid in the formulation of strategies that can assist day traders in making more rational decisions, ultimately leading to a reduction in future losses. This is particularly relevant in the context of volatile markets and the high-pressure environment in which day traders operate, where the ability to make informed decisions is crucial for minimizing capital losses.

The objective of this research paper is to determine the valence-arousal state of traders when they use limit orders and stop loss to their trades and compare their emotional states when these risk control measures are not applied. Our research question investigates whether there is a substantial difference between traders’ emotional states in the two scenarios. We propose a novel framework for multi-trader emotion classification that uses brainwave signals and deep learning algorithms and signal processing methods. We contribute to the electroencephalography-based emotion classification literature in financial trading by creating an EEG database of participants executing trades with and without stop loss and limit orders in two experiments to classify their emotional states. We recorded participants’ emotional state using a Self-assessment Manikin in a valence-arousal environment to answer our research question. The main aims of this research are:(1)To provide a technical framework for emotion classification in financial trading using EEG data and deep learning algorithms.(2)To determine the most appropriate neural network architecture and optimize it for improved EEG emotion classification and to achieve state-of-the-art accuracy.(3)To investigate the influence of limit orders and stop loss on traders’ emotional states and to generate an electroencephalography dataset for financial trading scenarios.

### 1.1. Contributions

The major contributions of this work are summarized as follows:This paper contributes to the field of emotion classification in financial trading by presenting an EEG database collected from 20 participants executing real-time trade and a framework for emotion classification utilizing EEG data and deep learning algorithms.A hybrid neural architecture is proposed and found to outperform previous studies.The study provides evidence that the use of risk control measures (limit orders and stop loss) has a significant impact on the emotional state of traders, and these measures can provoke affective states associated with trading.The empirical results demonstrate that the trading groups exhibit significantly different affective states from the control group.

### 1.2. Paper Structure

The remaining sections of the paper are organized as follows: Section 2 discusses the data collection procedure, data pre-processing, and the experimental design for the participants; Section 3 discusses the proposed framework and the deep learning-based emotion classification strategies utilized; Section 4 analyzes the empirical results; and Section 5 concludes the paper.

## 2. Dataset

### 2.1. Study Participants

We used electroencephalography (EEG) data in this study to validate the self-reported emotional states of twenty healthy individuals (*n* = 20, 10 males and 10 females, aged 25 to 60). All participants had prior stock trading experience to maintain the homogeneity of the group. Each participant was subjected to a single trial of two experiments of 30 min each. Prior to conducting the experiments, participants received an hour of hands-on training in the user interface of the simulated trading environment.

### 2.2. Data Acquisition and Recording Process

The EEG data were collected using the Interaxon Muse 2 headset. The Muse 2 EEG headset records brainwaves from four channels (AF7, AF8, TP9, and TP10) and a reference electrode (FPz) that acquires data from the Frontal (AF7 and AF8) and temporoparietal (TP9 and TP10) regions of the brain. Figure 1 depicts the Interaxon Muse 2 EEG headset used for recording brainwave signals. A new column was added to the dataset to label emotions at specified timestamps. The raw EEG signals recorded by the Muse 2 headband were transmitted from a mobile application, called Mind Monitor, to a Windows 10 machine using Open Sound Control streaming on port 5000 and saved in a flat file. To facilitate the downsampling of the data and generate a uniform stream frequency, all signals were recorded with a Unix timestamp. To minimize external noise and distractions, participants were not exposed to any other external noise or disturbances while conducting the study.

To ensure the integrity of the EEG signals, participants were instructed to minimize their movement and to keep their eyes open throughout the tasks. When looking at technical indicator charts, eye blink variability and lateral eye movement events can be associated with varied attentiveness levels; classification algorithms can account for these patterns of signal spikes [30]. The eye blink rate, which may affect AF7 and AF8 sensors, was not encouraged or discouraged to maintain a natural condition.

Throughout the 30-min experiment, participants were shown a Self-Assessment Manikin (SAM) [31] every minute and asked to rate their emotions on a scale of 1 to 10. The SAM is a tool that facilitates the rapid labeling of emotions without inherent bias and with less participant fatigue. These self-reported emotions were used as labels by the emotion classification algorithms. We used the digital version of SAM [32] in this study. Figure 2 shows the categorization of emotions that are captured according to the valence-arousal model via a SAM. Valence refers to the positive or negative dimension of emotion, while arousal refers to the degree or strength of the emotion. The combinations of valence and arousal can be transposed into several emotional states, namely High Valence High Arousal (HVHA), Low Valence High Arousal (LVHA), Low Valence Low Arousal (LVLA), and High Valence Low Arousal (HVLA). Hope, fear, regret, and calmness are the high-level emotions described by the valence-arousal framework, as depicted in Figure 2.

## 3. Proposed Methodology

This section describes the data pre-processing, feature extraction, emotion classification processes, and comparison of performance metrics with the benchmark dataset. The proposed framework for emotion classification consists of several essential steps to ensure high accuracy in emotion recognition. The first step involves the acquisition and recording of EEG signals along with the self-labeling of emotions using a Self-Assessment Manikin. Next, the acquired data undergoes a thorough cleaning and pre-processing stage described in Section 3.1. This step includes filtering and artifact removal at multiple levels to ensure that the data is high quality and free from noise and other disturbances that could affect the results. The pre-processed data undergoes feature extraction through the Independent Component Analysis (ICA) of various frequency bands, including gamma, beta, alpha, theta, and delta. This step is crucial as it helps to extract the most relevant information from the EEG signals for further analysis. Following feature extraction, the data is split into a training set and a testing set; the training set is then used to train various neural network models, including LSTM, CNN, CNN-LSTM, CNN-BiLSTM, and CNN-BiLSTM-CRF. Utilizing Bayesian Optimization, the hyperparameters of these classifiers are optimized to ensure their ideal performance. The final step in the proposed framework involves classifying emotions into the HVHA, LVHA, LVLA, and HVLA classes using the trained models. The performance of the models is then evaluated using various performance metrics, including accuracy, precision, sensitivity, specificity, F-score, ROC, and AUC, and compared with the previous literature. Figure 3 illustrates a self-explanatory flowchart of the overall experimental design approach.

### 3.1. Pre-Processing of EEG Data

We applied the following process for EEG data preprocessing and artifact removal. Figure 4 illustrates the steps involved in the data preprocessing procedure:Data collection: We imported the raw EEG data recorded from the Muse 2 headset, streamed it from the Mind Monitor application (mind-monitor.com, accessed on 22 November 2022), and saved it as a comma-separated file.Downsampling and data cleaning: The raw EEG data was recorded at 256 Hz, but we downsampled it to 128 Hz using EEGLab software, version 2022.1. To ensure high-quality data, we removed the first 30 s of data from the beginning of each participant’s signal which was designated for relaxation. We also removed non-informative components of the signal, such as missing data, non-numeric values, and variables related to the heart rate, accelerometer, and gyroscope that were collected but not studied.Low pass and high pass filtering: During EEG recording, the signals are interrupted by artifacts caused by eyeblinks, heart rate, and muscular movements such as jaw clenches [33]. We eliminated these artifacts from the resampled signal using two Butterworth IIR (Infinite Impulse Response) filters given by the following expression:
(1)H(s)=1/(1+(s/ωc)(2n))
where H(s) is the transfer function of the filter, s is the complex frequency variable, ωc is the cutoff frequency of the filter, and n is the order of the filter (i.e., the number of poles in the filter).

We used two Butterworth IIR filters to eliminate artifacts from the resampled signal caused by eyeblinks, heart rate, and muscular movements. A low-pass filter was applied at 45 Hz and then a high-pass filter at 1 Hz. Before filtering, we visually inspected the data and deleted any parts containing significant artifacts to minimize the spread of these artifacts during filtering. We only filtered continuous data segments and avoided filtering across boundaries to prevent filtering artifacts.

4.Notch filtering: To reduce noise from the 50 Hz frequency in the EEG signals, we used a notch filter to eliminate line noise originating from the electrical power supply. This step ensured that the EEG data was free from such noise and ready for further analysis.5.Channel rejection: We conducted checks to remove faulty channels by examining channels with no EEG activity for more than 5 s, channels with high noise (high standard deviation relative to other channels), and channels with low correlation with other channels (low correlation to other channels using a rejection threshold of 0.70).6.Automated Signal rejection (ASR) algorithm: We employed the ASR algorithm [34] for additional rejection of bad data segments based on the number of channels exceeding a 20 (default value) standard deviation threshold within a time window of 5 s. This allowed us to further reject portions of data that may have been missed by the earlier steps.7.Independent Component Analysis: We applied the ICA method [35] to further minimize the effects of artifacts such as ECG, EMG, EOG, and others that may still be present in the data despite multiple levels of cleaning. ICA is a successful method for dealing with artifacts such as muscle movements, eye blinks, lateral eye movements, heart rate, and others. However, to avoid strong artifacts affecting the data, we filtered them out in previous steps as ICA may not be able to remove them effectively.

Visual inspection: Finally, we removed any remaining artifacts manually using the visual inspection method in EEGLab with a sliding window of 30 s. Figure A1 and Figure A2 in the Appendix A show the EEG signals before and after visual inspection and artifact rejection.

### 3.2. Feature Extraction

After obtaining the cleaned EEG signals, we applied Independent Component Analysis (ICA) [35] on all four recorded channels to extract the features. ICA is a feature extraction method that decomposes a multivariate signal into independent components. In addition to the extraction of features, ICA also eliminates embedded artifacts that could be missed out in the manual artifact removal process. The ICA algorithm decomposes a non-stationary signal contained with several mixed frequencies into distinct, independent components (signals), each corresponding to a different frequency band. Each frequency band is associated with different emotional states. Table 1 shows the power spectrum of frequency bands that are associated with different emotional states [36]. The ICA method extracts gamma, beta, alpha, theta, and delta frequencies for each of the four channels. Consequently, each experiment with a single trial generated 20 features per dataset.

### 3.3. Emotion Classification Algorithms

This section discusses the neural network classifiers used for emotion classification following the input data’s successful preprocessing and feature extraction. To perform emotion classification, we used five different types of neural networks, namely, LSTM, CNN, CNN-LSTM, CNN-BiLSTM, and CNN-BiLSTM-CRF. Bayesian Optimization optimizes the hyperparameters of these classifiers.

#### 3.3.1. Long Short-Term Memory

Long Short-Term Memory (LSTM) is a type of recurrent neural network that is designed to model long-term dependencies in sequential data. Each LSTM cell takes in a sequence of input vectors $x_{1},x_{2},\ldots ,x_{T}$ and produces a corresponding sequence of output vectors $h_{1},h_{2},\ldots ,h_{T}$. The input vectors $x_{t}$ and output vectors $h_{t}$ at each time step t can have different shapes and dimensions depending on the specific task and dataset. Mathematically, the LSTM model can be represented by the below equations:(2)$$i_{t}=\sigma (W_{i}x_{t}+U_{i}h_{t-1}+b_{i})$$
(3)$$f_{t}=\sigma (W_{f}x_{t}+U_{f}h_{t-1}+b_{f})$$
(4)$$c_{t}=f_{t}c_{t-1}+i_{t}tanh(W_{c}x_{t}+U_{c}h_{t-1}+b_{c})$$
(5)$$o_{t}=\sigma (W_{o}x_{t}+U_{o}h_{t-1}+b_{o})$$
(6)$$h_{t}=o_{t}tanh(c_{t})$$
where $i_{t},f_{t},c_{t},$ and $o_{t}$ are intermediate variables; $h_{t}$ represents the output of the LSTM cell at time step t; $x_{t}$ and $h_{t-1}$ are the input data and previous hidden state at time step t, respectively; U and W are weight matrices, and b is a bias vector. The sigmoid function is represented by σ, and the hyperbolic tangent function is represented by tanh.

#### 3.3.2. Convolutional Neural Network

A Convolutional Neural Network (CNN) consists of three layers, namely, a convolutional layer, a pooling layer, and a fully connected layer [37]. The output of every layer is connected to a small neighborhood in the input through a weight matrix also known as a filter or kernel. Each filter is moved around the input giving rise to one 2D output. The outputs corresponding to each filter are stacked, giving rise to an output volume. The convolution operation of a 2D input signal x(p,q) can be expressed as follows:(7)$$y(m,n)=\sum_{p=0}^{\infty }\;\sum_{q=0}^{\infty }\;x(p,q)h(m-p,n-q)$$
where y(m,n) is the output obtained after the convolution operation and h is the kernel coefficient of the system.

Following a series of convolutions, there is a pooling layer that is used to reduce the dimensionality of feature vectors. The main job of a pooling layer is to provide translational invariance by subsampling the input feature vector. The pooling layers are followed by a fully connected layer that acts as a classifier. Figure 5 shows the basic structure of a CNN model.

#### 3.3.3. CNN-LSTM

A CNN-LSTM model integrates the CNN model for feature selection with a Long Short-Term Memory (LSTM) model for time series prediction. A convolutional layer in this model identifies the important features in the input data [38]. Pooling reduces the data dimensions. The pooling layer interfaces with LSTM and linked layers (also known as dense layers). The combined CNN-LSTM model can be expressed as follows:(8)$$h_{t}=LSTM(x_{t},h_{t-1})$$
(9)$$y_{t}=CNN(h_{t})$$
where $x_{t}$ represents the input data at time t, $h_{t-1}$ represents the hidden state of the LSTM at time t−1, and $h_{t}$ represents the hidden state of the LSTM at time *t*.

#### 3.3.4. CNN-BiLSTM

A CNN-BiLSTM [39] model incorporates the CNN for feature selection and the bidirectional LSTM (BiLSTM) model for time series forecasting. A BiLSTM model has an auxiliary LSTM layer that reverses the information flow. As the direction of the input sequence is reversed due to the additional LSTM layer, the model can consider incoming information from both forward and reverse directions. Figure 6 shows a schematic structure of a CNN model combined with an LSTM or BiLSTM model.

The CNN-BiLSTM model can be expressed as follows:(10)$$h_{t}^{f}=LSTM(x_{t},h_{t-1}^{f})$$
(11)$$h_{t}^{b}=LSTM(x_{t},h_{t+1}^{b})$$
(12)$$y_{t}=CNN([h_{t}^{f},h_{t}^{b}])$$
where $x_{t}$ represents the input data at time t, $h_{t-1}^{f}$ and $h_{t+1}^{b}$ represent the forward and backward hidden states of the BiLSTM at time t−1 and t+1, respectively, and $h_{t}^{f}$ and $h_{t}^{b}$ represent the forward and backward hidden states of the BiLSTM at time t. The function LSTM represents the LSTM network, which processes the input data $x_{t}$ and hidden state $h_{t-1}^{f}$ or $h_{t+1}^{b}$ to produce the hidden state $h_{t}^{f}$ or $h_{t}^{b}$. The function CNN represents the CNN network, which processes the concatenated hidden states $[h_{t}^{f},h_{t}^{b}]y_{t}$ to produce the output $y_{t}$.

#### 3.3.5. CNN-BiLSTM with Conditional Random Field

Conditional Random Field (CRF) is a type of probabilistic graphical model that can be used to model the relationships between different variables in a dataset. EEG data contains complicated temporal correlations between the data and labels as a result of fluctuating emotional states in response to price changes and losing or winning a transaction, therefore, establishing the appropriate label sequence can be a challenging task. The addition of a CRF layer facilitates the modeling of these dependencies and thus helps to improve the accuracy of predictions in a CNN-BiLSTM model for the classification of emotions from the EEG data. The CRF layer [40] can be defined as follows:

Let $y=\left(y_{1},\ldots ,y_{T}\right)$ be the sequence of labels and $x=\left(x_{1},\ldots ,x_{T}\right)$ be the sequence of input features. The CRF layer computes the conditional probability of the label sequence given the input features as follows:(13)$$p(y|x)=\frac{1}{Z(x)}\prod_{t=1}^{T}\;\psi_{t}\left(y_{t},y_{t-1},x\right)$$
where Z(x) is the normalization factor, also known as the partition function, and $\psi_{t}$ is the transition score from label $y_{t}$ to label $y_{t-1}$ at time step t.

The transition scores are defined as:(14)$$\psi_{t}\left(y_{t},y_{t-1},x\right)=exp\left(\sum_{k=1}^{K}\;tr\left(k,y_{t},y_{t-1}\right)\cdot emit\left(k,y_{t},x_{t}\right)\right)$$
where K is the number of classes, $tr\left(k,y_{t},y_{t-1}\right)$ is the transition weight from class k at time step t−1 to class k at time step t, and $emit\left(k,y_{t},x_{t}\right)$ is the emission weight from class k at time step t to the input features $x_{t}$.

#### 3.3.6. Hyperparameter Optimization

We applied the Bayesian Optimization [41] approach to find the best possible values for the hyperparameters of the specified models. Bayesian Optimization is well-suited for tackling black-box optimizations and noisy functional evaluations, and it is more efficient than manual network-tuning or traditional approaches such as the Grid-Search and Randomized-Search in high-dimensional datasets. Table A1 in Appendix A shows the search space employed by the Bayesian Optimization technique for all classifiers. The optimization process considers the learning rate, dropout rate, count of hidden layers, neuron count per layer, batch size, activation function, optimizer, number of CNN filters, kernel size, pooling size, and type, among other hyperparameters.

In this paper, we used the same search space configuration for both experiments. To optimize the hyperparameters, we utilized the Adaptive Experimentation (AE) library [42] in Python programming language. AE is a widely adopted multi-objective optimization framework that has been employed effectively by Facebook in numerous real-world online experiments.

### 3.4. Performance Metrics and Experimental Software

In this research, accuracy, precision, recall (sensitivity), specificity, and negative predicted values from the confusion matrix are used as the performance metrics for classification algorithms.

Accuracy is the ratio of the number of true positives and true negatives to the total number of predictions made by the model. It is given by the below expression:(15)$$Accuracy=\frac{TP+TN}{TP+TN+FP+FN}$$

Precision refers to the proportion of correctly classified positive instances classified as positive by the classifier. Mathematically, it is given by:(16)Precision =TP/(TP+TN)

Recall (Sensitivity, True Positive Rate) is the proportion of positive instances that were correctly classified as positive by the classifier. The recall is given by:(17)Recall =TP/(TP+FN)

The NPV (Negative Predicted Value) measures the proportion of negative instances that were correctly classified as negative by the classifier. NPV is expressed as follows:(18)NPV=TN/(TN+FN)

The F1-Measure represents the harmonic mean of precision and recall. F1-Measure provides a comprehensive evaluation of the classifier’s performance considering both precision and recall. It is given by:(19)F1−Measure =2× (Precision × Recall)/(Precision + Recall)
where TP = True Positives (number of instances correctly classified as positive), TN = True Negatives (number of instances correctly classified as negative), FP = False Positives (number of instances incorrectly classified as positive), and FN = False Negatives (number of instances incorrectly classified as negative).

## 4. Experiments

The proposed emotion classification framework compares participants’ emotional state when trading with predetermined risk locking to their emotional state when trading without risk locking. Each experiment was simulated using live Bitcoin market data with 1-min interval candlestick charts for a duration of 30 min. In the experiments, participants were given a starting balance of USD 10,000 for use in simulated trades. The initial trading amount was uniform across all participants and was not adjusted based on individual characteristics or prior trading experience. The initial amount was set at a relatively large sum in order to give participants the opportunity to experience a wide range of potential outcomes and to allow for the analysis of their emotional responses under different trading actions (buy, hold, and sell).

In the first experiment, participants were given a set of predetermined risk-locking measures to use while making simulated trades. These measures included a pre-defined stop loss of 1% of the traded value and a 1% purchase limit order of the traded value, which were set based on the anticipated 30-min price volatility. A stop loss is a risk management tool that automatically closes a trade when the loss reaches a predetermined price threshold, helping to prevent the trader from incurring further losses. A buy limit order is an order to buy a security at a specified price or lower. It is automatically executed when the price reaches the upper limit of the specified threshold. These risk-locking measures were intended to help participants manage their potential losses and make informed trading decisions.

In contrast, the second experiment did not have any predetermined buy limit orders or stop loss measures to lock in risks. Participants were given the freedom to make trading decisions based on their own assessment of the market and to profit from fluctuations in Bitcoin’s price by buying, holding, or selling their holdings.

A control group was employed in this study to establish a baseline for comparison with the trading groups. The control group refrained from any trading activity and was not provided with the options of buy, sell, or hold actions. Instead, the affective states of the control group were derived solely from observing the market data. We aimed to discern any significant distinctions in emotional responses that could be attributed to trading activity by comparing the affective states of the control group with those of the trading groups.

The subsequent section describes how each electrode’s raw EEG data was converted into brainwaves (delta, theta, alpha, beta, and gamma frequencies) which were then categorized into different emotional states using machine learning algorithms.

All experiments were executed on a Windows 10 operating system, equipped with an Intel Core i5 processor with 8 cores operating at a frequency of 2.30 GHz, 20 GB of RAM, and an NVIDIA GeForce GTX 1050 graphics processing unit. The data preprocessing and deep learning techniques were implemented utilizing Python 3.7.3 (64-bit), TensorFlow 2.6, and CUDNN 10.1.105. The EEG analysis was performed utilizing the MNE library 1.3.0 (https://mne.tools, accessed on 23 December 2022) and Matlab’s EEGLab toolbox.

## 5. Results and Analysis

### 5.1. Network Architecture Suggested by Bayesian Optimization

Table 2 presents the suggested network architecture for both Experiment 1 and Experiment 2 based on the hyperparameter values obtained using Bayesian Optimization. We utilized these hyperparameter values for the deep learning models used for emotion classification. The CNN portion of the models used 128 CNN filters with a kernel size of 3, a pooling layer with a size of 2, and a max pooling type as obtained from Bayesian Optimization.

### 5.2. Distribution of Valence-Arousal for Experimental Groups

Figure 7 presents the distribution of valence-arousal for both experimental groups. In both experiments, the most dominant emotion is High Valence High Arousal (HVHA), followed by Low Valence Low Arousal (LVLA), Low Valence High Arousal (LVHA), and High Valence Low Arousal (HVLA). Upon further analysis, it can be observed that while the overall emotion experienced by participants in both experiments is similar, there is a notable difference in the distribution of emotions experienced under different trading strategies. Specifically, Experiment 1, which employed predetermined stop loss and limit orders to lock in profit and risk objectives, exhibited a higher frequency of HVHA and LVLA compared to Experiment 2, which did not utilize such risk-locking techniques.

### 5.3. Effect of Stop Loss and Limit Orders on Participant Emotions

The results of Experiment 1 showed that the use of stop loss and limit orders on trades was associated with a higher High Valence High Arousal (HVHA) emotion (56%) compared to the group in Experiment 2, which did not use these risk-locking criteria (46%). In other words, the test group that utilized stop loss and limit orders was 17.85% more hopeful than the group that did not use these criteria. The frequency of Low Valence Low Arousal (LVLA) emotions was similar between the two groups, although there was a marginally higher frequency in Experiment 1.

One of the most notable findings of this study was that the use of stop loss and limit orders were associated with a lower frequency of Low Valence High Arousal (LVHA, or fear) emotions. Specifically, participants in Experiment 1 who employed these risk-reward control measures reported LVHA emotions of 13%, compared to 26% in the group that traded without these measures in Experiment 2. This difference can be attributed to the ability of stop loss and rules to minimize the impact of volatility and significant losses [22]. Consequently, traders experience less fear when entering or exiting positions due to these risk control methods.

Another notable finding was the absence of High Valence Low Arousal (HVLA, or calmness) emotions in the second group. This may be expected, as it is rare for traders to exhibit calmness during highly volatile short-term trading sessions. In comparison, the frequency of HVLA emotions was low but detectable in the first group. Overall, these results suggest that the use of stop loss and limit orders is associated with higher frequencies of HVHA and lower frequencies of LVHA emotions, indicating that managing risks in this way may lead to more hopeful and less fearful traders.

### 5.4. Analysis of Classification Accuracies in Experiments 1 and 2

The classification accuracies for each participant in Experiment 1 and Experiment 2 are presented in Table 3 and Table 4, respectively. The classification accuracy of each participant is dependent on variances in their physiological features and the intensity or complexity of their conveyed emotions. The results of the ablation study demonstrate the mean classification accuracy for each participant using a 10-fold cross-validation technique for both Experiments 1 and 2. Results from Experiment 1 show that the CNN-BiLSTM-CRF model achieved the highest mean accuracy of 85.65%, followed by the CNN-BiLSTM, CNN-LSTM, and CNN models. In Experiment 2, the highest mean accuracy of 85.05% was also obtained by the CNN-BiLSTM-CRF model, with similar performance for the CNN-BiLSTM, CNN-LSTM, and CNN models. The results of this study suggest that the incorporation of a CRF layer in the CNN-BiLSTM architecture improves the overall accuracy of emotion classification.

Furthermore, it can be observed from Figure 8 that there is a clear progression in terms of accuracy as the model complexity increases, with LSTM achieving the lowest accuracy and CNN-BiLSTM-CRF achieving the highest accuracy. The CNN-BiLSTM-CRF model achieved the highest mean accuracy with the least deviation in performance, unlike LSTM and the CNN models which exhibited high variability in their performance. The CNN-BiLSTM-CRF model produced the highest mean accuracy with the least performance variance in contrast to other competitive models. This can be attributed to the use of a CRF layer which helps to regularize the model and reduce overfitting, resulting in more consistent and reliable performance. Additionally, it allows the model to consider the context of previously predicted classes, thus enforcing smoothness constraints on the predicted sequence and aligning predictions from different parts of the sequence. This can help mitigate the effect of outliers in the data.

### 5.5. Influence of Non-Systematic Factors and Subject Level Differences

In this study, an exhaustive methodology was applied to preprocess the EEG data for all participants. However, it is possible that other non-systematic factors may have affected the EEG signals of some participants. Therefore, the reported accuracy differs on an individual basis. Compared to other participants, subjects 1, 3, 11, and 12 reported marginally inferior accuracy in both experiments. Further investigation using spectral analysis of the EEG data for these participants may be necessary to understand this phenomenon.

### 5.6. Comparison of Precision, Recall, F1-Measure, Specificity, and NPV

Table 5 and Table 6 present the mean classification performance metrics of EEG emotion classification models in Experiments 1 and 2. In Experiment 1, the model with the highest performance across all metrics (precision, recall, F1-measure, specificity, and NPV) is CNN-BiLSTM-CRF. Similarly, in Experiment 2, the performance metrics for various models reveal that CNN-BiLSTM-CRF is the most robust model. The results show that the addition of a Conditional Random Field (CRF) layer to the CNN-BiLSTM model improves the overall performance of the model by capturing the dependencies between the output labels and providing more accurate predictions.

### 5.7. ROC Analysis of Cross-Validated Emotion Classification Performance

Figure 9 and Figure 10 depict the mean cross-validated Receiver Operating Characteristic (ROC) curves for the training and test data in Experiments 1 and 2, respectively. The ROC curve helps analyze the performance of binary classifiers and exhibits the relationship between the True Positive Rate (TPR) and False Positive Rate (FPR) (FPR). The Area Under Curve (AUC) is used to evaluate the overall performance of a classifier, with a larger AUC value indicating a superior classifier. In Figure 9 and Figure 10, the cross-validated ROC curves of the models are displayed for the training and test data in Experiment 1 and Experiment 2, respectively. A thorough analysis of the figures highlights the robustness of the CNN-BiLSTM-CRF model, as it consistently achieves a significantly higher Area Under Curve (AUC) than the other models across all four classes in both experiments. This result suggests that the proposed model performs better in accurately classifying emotions.

### 5.8. Contribution of EEG Features in the Proposed Model

In this section, we discuss the results of feature importance obtained for Experiment 1, Experiment 2, and the control group which performed no trading activity.

The results of the deepSHAP algorithm’s feature importance analysis in Experiments 1 and 2 are shown in Figure 11a,b. The results in Experiment 1 indicate that the most significant features, in order of importance, were gamma_Af7, gamma_AF8, gamma_TP9, gamma_TP10, beta_AF8, alpha_AF7, delta_TP9, beta_TP10, and alpha_AF8. Out of the total 20 features present in the EEG dataset, the dominance of gamma-related features can be linked to their association with problem-solving, concentration, and positive valence, as well as the increased arousal that typically results from high-intensity feelings dominated by the outcomes of win or loss. Conversely, Theta-related features, which are associated with deep relaxation and REM sleep, were the least significant and can be closely tied to the HVLA emotion class. This is supported by the distributions of valence-arousal shown in Figure 7, which suggest that, despite having trading experience, the participants in the study did not exhibit a significant amount of HVLA emotion.

The feature importance analysis in Experiment 2, shown in Figure 11b, shows that beta-related features are the most dominant followed by gamma and alpha-related features, while theta and delta-related features are the least significant. According to Table 1, Beta related features are closely associated with concentration and decision-making. Figure 12a,b shows the proposed model’s local feature contributions. The local approach offers a deeper understanding of the relationship between individual EEG features and their impact on class predictions for individual instances. The feature importance for specific instances is consistent with the results of the global method, with gamma and beta-related features remaining the most important features and theta and delta being less significant.

The global and local feature importance for the control group is shown in Figure 13a,b. The participants in this group simply observed the market data without performing any actual trading activity. This group serves as a baseline for comparison with Experiment 1 and Experiment 2 which involved active trading activity. The global and local feature importance analysis of the control group shows that the Delta and Theta-related features are the most dominant features along with a minor contribution from Alpha AF8 (global) and Alpha AF7 (local). According to Table 1, Delta and Theta are associated with peaceful states and resting states. This state can be most closely linked to HVLA (calmness) as shown in Figure 2. This contrasts with the dominant features observed in Experiment 1 and Experiment 2 where gamma, beta, and alpha are associated with problem-solving, concentrating, and actively thoughtful states. Moreover, the results indicate that HVLA was negligible in Experiment 1 and absent in Experiment 2, which suggests that even experienced traders find it challenging to remain calm during the event of a win or loss. However, when they are not actively engaged in trading, HVLA becomes dominant. The difference in the feature importance between the control and trading groups can be attributed to the effect of trading activity.

### 5.9. Comparison with Previous Works

In addition to the ablation experiments, we also compared our research results to those of other closely related studies. These studies employed similar self-labeling and emotion identification methods as our own but used different experimental scenarios for financial trading. Table 7 presents this comparison of the classification performance of the methodology proposed in this paper and earlier EEG-based emotion recognition studies in financial trading. Table 7 shows that the CNN-BiLSTM-CRF classifier achieved a superior accuracy of 85.65% compared to previous works. In the context of similar multi-class emotion categorization studies, it can be inferred that the proposed methodology is both effective and competitive.

## 6. Discussion

The present study aimed to determine the emotional state of traders when utilizing limit orders and stop loss in their trades and compare it to when these risk control measures were not applied. A novel framework for multi-trader emotion classification using EEG signals and deep learning algorithms was proposed and utilized. The study made several important contributions to the field of emotion classification in financial trading. Firstly, a new EEG database was created and the proposed framework was shown to outperform previous studies, achieving state-of-the-art accuracy levels of 85.65% and 85.05% in two experiments. The model was robust with regard to artifacts, with multiple levels of artifact removal implemented, and its performance was evaluated across multiple performance metrics. Furthermore, feature contribution was explained using deepSHAP. The Self-assessment Manikin was used for self-labeling, providing a more reliable and accurate measurement of emotional states. The results demonstrated an association between the application of stop loss and limit orders as risk management measures and a reduction in the frequency of fear (measured by LVHA) among traders. This decrease in fear was linked to the effectiveness of stop loss and limit orders to offset the effects of market volatility, prospective losses, and significant gains. One limitation of this study is the exclusive use of a homogeneous group of experienced participants with prior trading experience. Future research may broaden the EEG data sample by incorporating participants from a broader range of backgrounds, including novice traders.

Table 8 summarizes various studies by presenting the models used, the feature extraction methods employed, the number of participants involved, and the advantages and disadvantages of each study. This comparison highlights each study’s strengths and weaknesses and serves as a baseline for the performance of the current study and future works in this area.

## 7. Limitations and Future Work Directions

While the findings of this study are promising for modulating the affective states of participants, it is essential to acknowledge and consider the limitations in the context of future research in this area.

Emotion is a complex subconscious process regulated by the amygdala in the brain [44]. This study uses the participants’ self-reported feelings as a measure of emotion by employing the digital variant [32] of SAM [45] which is one of the most accepted measurement tools for human emotions. However, self-reported feelings characterized by valence and arousal provide a limited understanding of the complexity of emotions. Therefore, future studies should incorporate additional physiological features such as cardiovascular (heart rate, heart rate variability, and inter-beat intervals), electrodermal, and respiratory data, among others as suggested by [9]. This analysis may aid in further enhancing the understanding of the emotional elicitation process during trading activities.

Secondly, it is important to note that EEG signals are susceptible to various external factors such as noise, incorrect electrode placement, and environmental conditions. In addition, participants’ physical and mental states can be influenced by individual biases, personal factors, mood, and past experiences, which can result in variations in their emotional responses. A small change in any of these factors can lead to signal distortion or changes in emotional response. While our study employed rigorous preprocessing methods to mitigate the effects of external factors on EEG signals, it is important to acknowledge that these factors can never be fully eliminated. Thus, future studies may benefit from continued efforts to optimize data collection and preprocessing techniques.

Third, the participant group in this study is a reasonably homogeneous group of traders to replicate everyday work activities in a competitive environment. However, future studies could investigate more heterogeneous participant groups to obtain more generalizable results and broaden the scope of our work.

## 8. Conclusions

The main objective of this study was to investigate the effect of using stop loss and limit orders on the emotional state of traders during short-term trading sessions. A deep learning-based valence-arousal framework for emotion classification was developed and applied to EEG data collected from 20 participants who participated in two single-trial experiments. The first group of participants was required to trade with stop loss and limit orders, while the second group was instructed to trade without using these risk control measures. The participants self-labeled their emotional states using a Self-assessment Manikin.

We proposed a hybrid neural architecture combining a Conditional Random Field layer with a CNN-BiLSTM model and used Bayesian Optimization to determine the optimal hyperparameters for multi-trader emotion classification. The results showed that the CNN-BiLSTM-CRF model achieved the highest mean accuracy and least deviation in performance. The model outperformed the previous literature in EEG-based multi-class emotion recognition. Our findings revealed that the group trading with stop loss and limit orders exhibited a greater HVHA (more hope) and a lower LVHA (less fear). In addition, HVLA (emotion of calmness) was found to be insignificant regardless of whether traders applied risk control via limit orders and stop loss or not.

The findings of this study have substantial implications for day traders and portfolio managers. The results of this research can help traders better understand their emotional state and develop positive decision-making skills and the EEG dataset generated can be used for future behavioral finance studies focusing on financial trading. In future research, we plan to expand our EEG dataset through additional testing on a larger sample size and more diverse population in order to gain a deeper understanding of the emotional states in financial trading scenarios. We also intend to explore the applicability of the proposed emotional classification architecture to other domains.

## Figures and Tables

**Figure 1 sensors-23-03474-f001:**
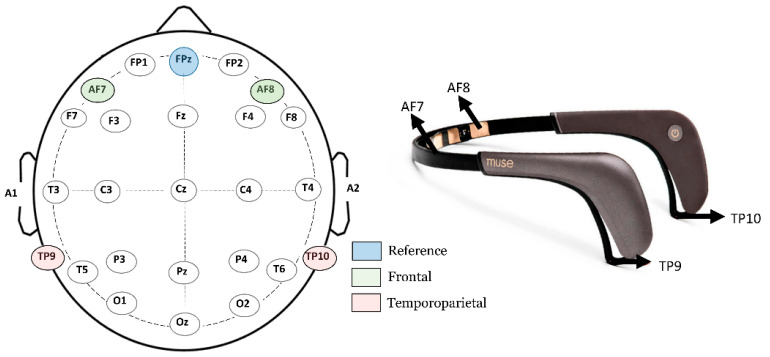
The four channels AF7, AF8, TP9, and TP10, highlighted in the 10–20 electrode placement scheme, and the EEG headband utilized to acquire EEG data.

**Figure 2 sensors-23-03474-f002:**
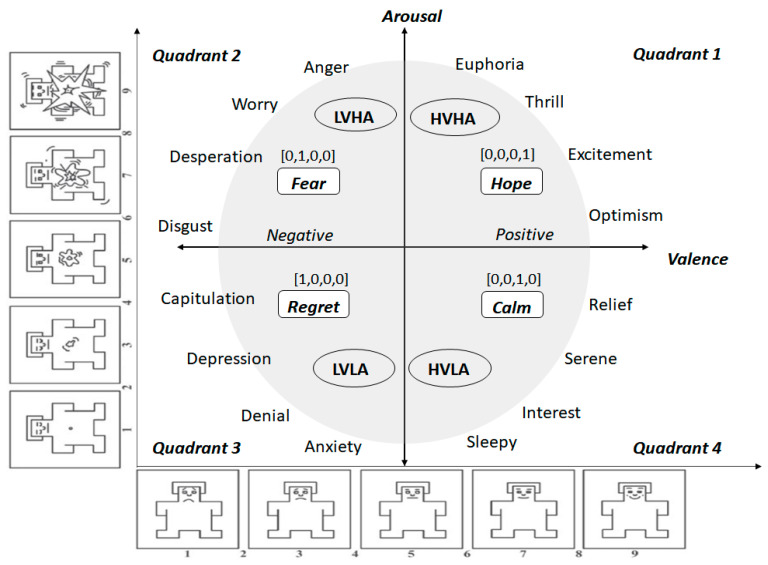
A schematic illustration of the self-labeling of emotions using SAM within the framework of valence and arousal.

**Figure 3 sensors-23-03474-f003:**
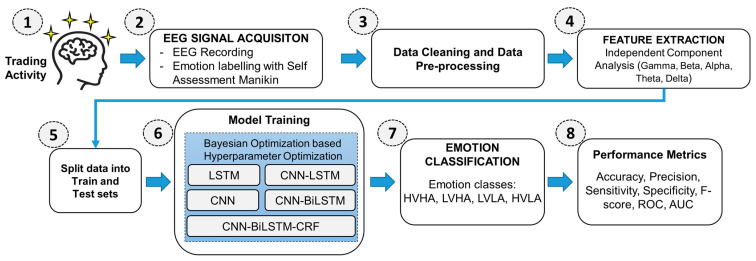
Workflow schematic of the proposed emotion classification framework.

**Figure 4 sensors-23-03474-f004:**
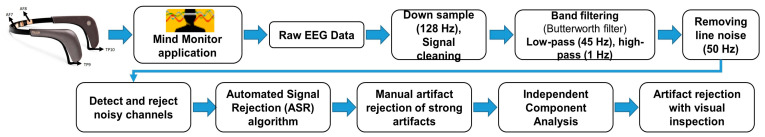
Schematic Illustration of data pre-processing for artifact removal and signal denoising.

**Figure 5 sensors-23-03474-f005:**
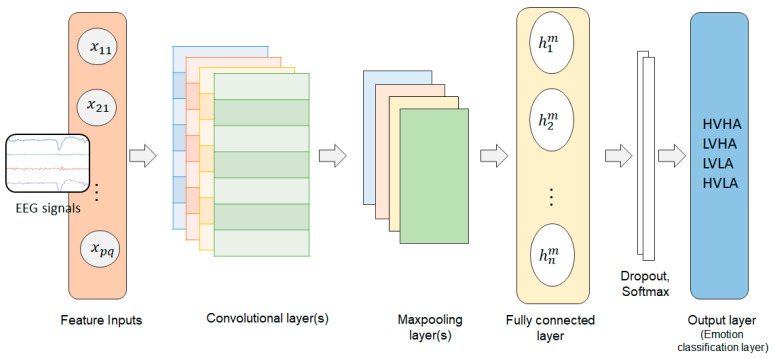
Block diagram of a convolutional neural network.

**Figure 6 sensors-23-03474-f006:**
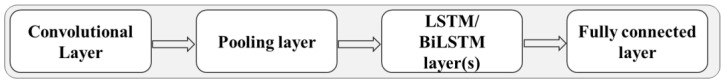
CNN structure depicting the interface between the pooling layer and an LSTM or BiLSTM.

**Figure 7 sensors-23-03474-f007:**
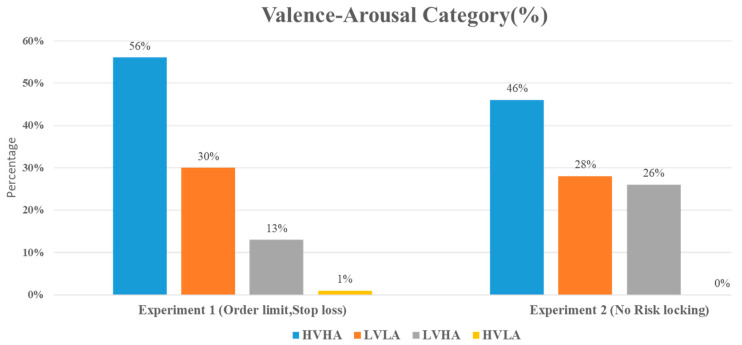
The percentage of valence-arousal category distribution obtained for both experiments conducted with and without risk locking via limit order and stop loss. The second group, which did not use stop loss and limit orders to lock in the risks and book the profits, demonstrated twice (26%) as much Low Valence and High Arousal (the emotion of fear) as the first group (13%), which did utilize these measures.

**Figure 8 sensors-23-03474-f008:**
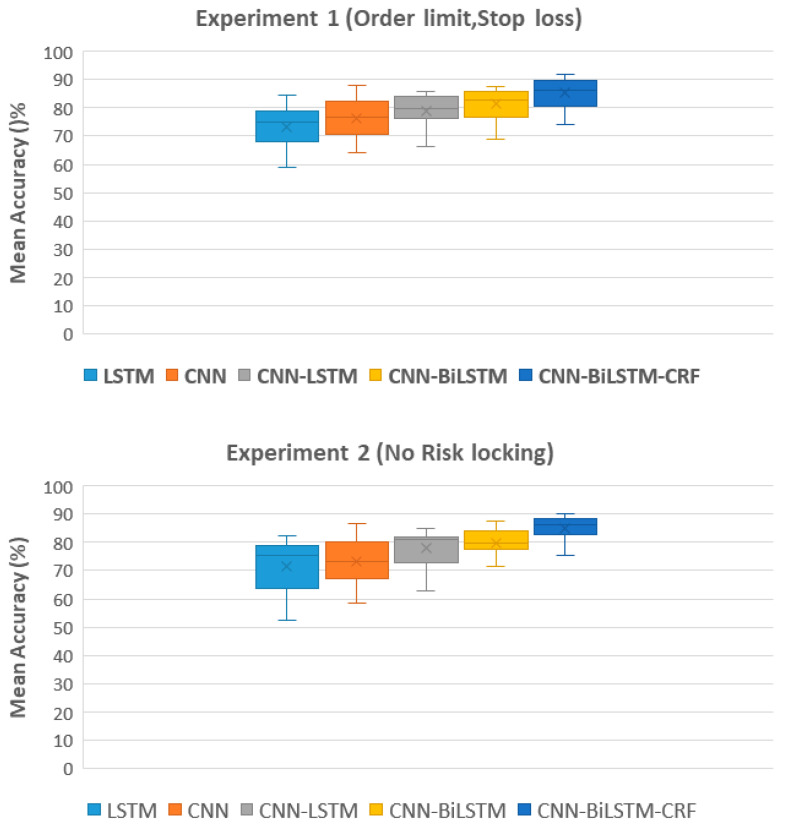
Comparison of the mean classification accuracy across different models for Experiment 1 with risk locking and Experiment 2 without risk locking, as determined by a 10-fold cross-validation technique.

**Figure 9 sensors-23-03474-f009:**
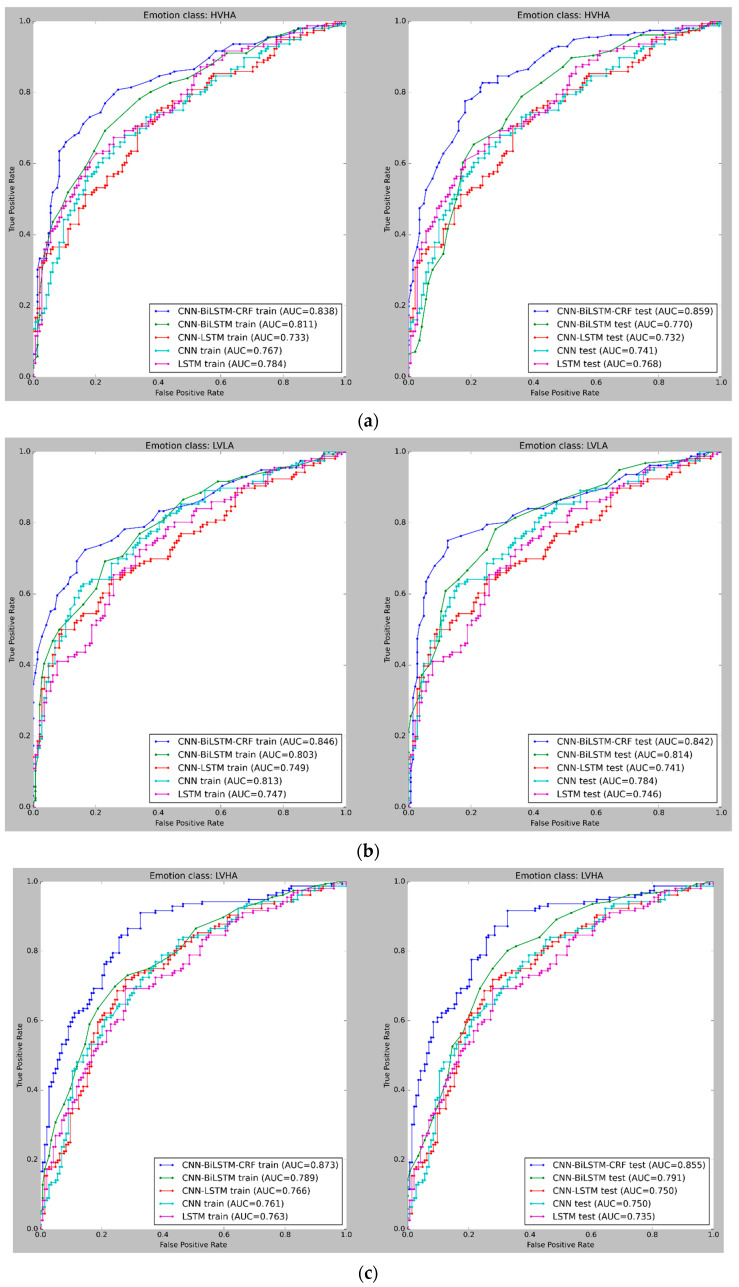
(**a**) ROC curves for emotion classification in Experiment 1, using cross-validated training and test datasets for High Valence High Arousal (HVHA). The curve shift towards the upper-left quadrant indicates improved model performance. (**b**) ROC curves for emotion classification in Experiment 1, using cross-validated training and test datasets for Low Valence Low Arousal (LVLA) class. (**c**) ROC curves for emotion classification in Experiment 1, using cross-validated training and test datasets for Low Valence High Arousal (LVHA) class. (**d**) ROC curves for emotion classification in Experiment 1, using cross-validated training and test datasets for High Valence Low Arousal (HVLA) class.

**Figure 10 sensors-23-03474-f010:**
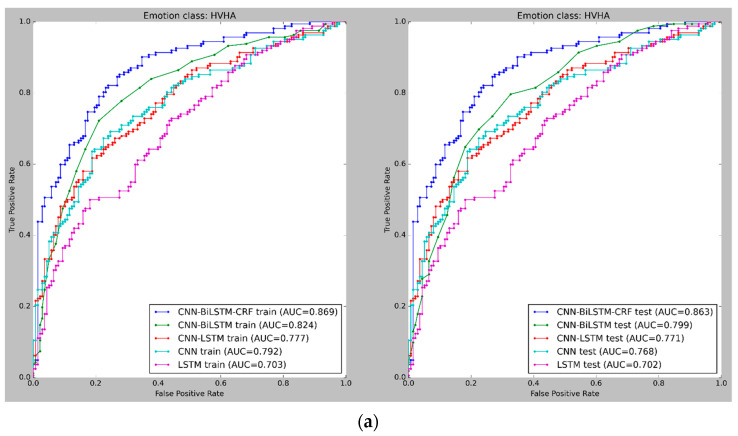
(**a**–**c**) ROC curves for emotion classification in Experiment 2, using cross-validated training and test datasets for all four emotion classes: High Valence High Arousal (HVHA), Low Valence Low Arousal (LVLA), and Low Valence High Arousal (LVHA). (**d**) ROC curves for emotion classification in Experiment 2, using cross-validated training and test datasets for all four emotion classes: High Valence Low Arousal (LVLA).

**Figure 11 sensors-23-03474-f011:**
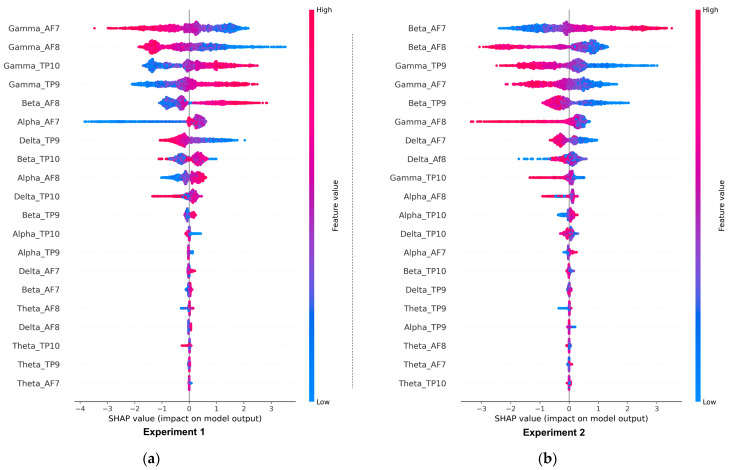
(**a**,**b**) Global feature importance of various EEG features in the proposed model (CNN-BiLSTM-CRF) for Experiment 1 and Experiment 2, respectively.

**Figure 12 sensors-23-03474-f012:**
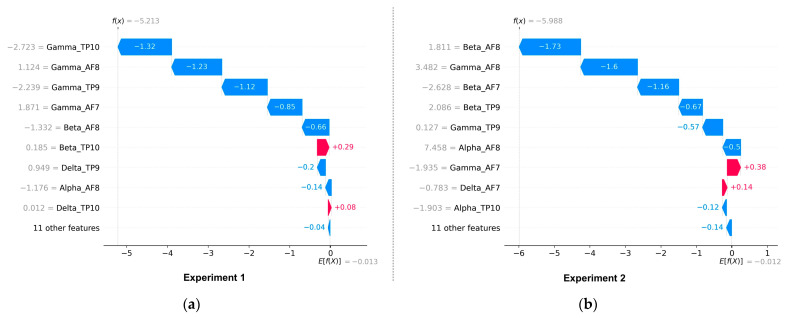
(**a**,**b**) Local contribution of various EEG features in the proposed model (CNN-BiLSTM-CRF) for Experiment 1 and Experiment 2, respectively.

**Figure 13 sensors-23-03474-f013:**
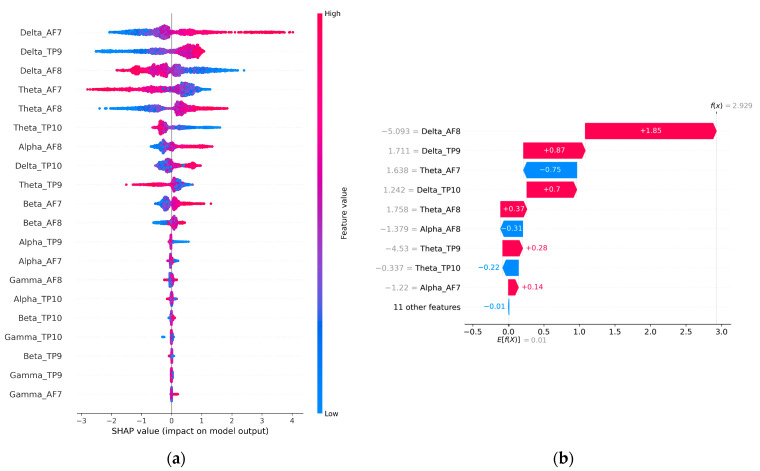
(**a**,**b**) Global and local contribution of various EEG features in the proposed model (CNN-BiLSTM-CRF) for the control group.

**Table 1 sensors-23-03474-t001:** Emotional state association of brainwaves at different frequency bands.

Frequency Band	Frequency Range	Emotional State Association
Gamma	>30 Hz	Problem-solving, concentrating, associated with positive valence. Arousal increases with high-intensity stimuli
Beta	13–20 Hz	Awake, busy, normal activities, and alert
Alpha	8–13 Hz	Relaxed, calm, and reflective
Theta	4–7 Hz	Deep relaxation
Delta	0–4 Hz	REM sleep and dreaming

**Table 2 sensors-23-03474-t002:** The suggested network architecture based on the hyperparameter values obtained using Bayesian Optimization for both experiments.

Experiment	Model	Dropout Rate	Learning Rate	Hidden Layers	Neurons per Layer	Activation Function	Batch Size	Optimizer
Experiment 1	LSTM	0.022	0.0001	3	461	linear	32	adam
Experiment 1	CNN	0.262	0.001	3	463	relu	8	adamx
Experiment 1	CNN-LSTM	0.045	0.0010	2	450	relu	8	adam
Experiment 1	CNN-BiLSTM	0.220	0.0310	2	515	relu	32	adam
Experiment 1	CNN-BiLSTM-CRF	0.210	0.0128	3	535	relu	64	adam
Experiment 2	LSTM	0.025	0.0105	3	444	linear	64	adamx
Experiment 2	CNN	0.202	0.0610	3	401	relu	4	adam
Experiment 2	CNN-LSTM	0.039	0.0120	2	494	relu	8	adamx
Experiment 2	CNN-BiLSTM	0.210	0.0262	2	463	relu	8	adam
Experiment 2	CNN-BiLSTM-CRF	0.097	0.0121	3	470	relu	32	adamx

**Table 3 sensors-23-03474-t003:** Mean classification accuracy with a 10-fold cross-validation technique for each participant in Experiment 1. Ten test runs were performed on each model, and the average of the results was calculated.

Participant	LSTM	CNN	CNN-LSTM	CNN-BiLSTM	CNN-BiLSTM-CRF
1	59.06	64.03	66.43	75.72	78.77
2	79.16	84.49	84.56	85.68	89.91
3	61.11	67.58	69.30	68.97	76.90
4	78.83	79.32	80.47	85.92	86.12
5	69.24	72.90	77.66	81.58	85.98
6	84.49	88.03	85.16	86.82	89.79
7	75.01	80.35	85.82	85.59	89.63
8	80.77	85.25	84.47	87.52	89.54
9	74.69	76.38	78.00	79.33	84.31
10	73.03	76.63	76.48	82.69	84.90
11	63.90	70.60	73.53	74.19	79.90
12	62.66	65.54	70.88	70.58	79.88
13	67.52	80.22	81.52	79.04	86.19
14	72.61	75.18	79.73	85.71	88.84
15	83.41	84.46	85.57	87.55	89.71
16	75.10	75.85	81.55	82.53	86.82
17	80.87	82.75	82.98	85.46	89.89
18	77.65	77.80	79.13	75.98	83.89
19	74.90	71.58	76.35	82.35	84.75
20	70.80	65.72	79.36	86.62	87.21
**Mean Accuracy**	**73.24**	**76.23**	**78.95**	**81.49**	**85.65 ****

The numbers in bold represent the mean accuracy; ****** denotes the highest accuracy reported.

**Table 4 sensors-23-03474-t004:** Mean classification accuracy for each participant in Experiment 1 using a 10-fold cross-validation technique. Ten test runs were performed on each model, and the average of the results was calculated.

Participant	LSTM	CNN	CNN-LSTM	CNN-BiLSTM	CNN-BiLSTM-CRF
1	52.42	65.44	62.97	73.09	76.88
2	82.25	84.36	80.92	81.42	90.10
3	57.14	59.63	70.03	71.56	75.62
4	80.25	83.39	84.95	87.43	86.10
5	63.21	68.15	69.67	80.32	83.96
6	79.13	80.88	81.03	86.10	88.80
7	68.52	73.08	81.87	84.28	87.72
8	74.50	84.23	81.20	84.28	87.79
9	69.50	76.64	75.79	78.51	82.92
10	76.59	74.44	81.17	84.26	88.73
11	61.73	65.19	72.65	74.53	81.87
12	59.84	66.78	72.61	73.01	80.49
13	65.39	75.70	82.89	79.36	82.88
14	75.63	72.25	73.62	77.38	84.19
15	78.53	78.82	81.83	84.13	89.59
16	76.08	68.74	83.71	78.66	86.84
17	82.35	86.80	83.72	78.65	88.45
18	75.59	73.21	80.88	77.57	83.57
19	79.53	72.31	74.39	82.81	87.95
20	75.02	58.71	81.40	81.35	86.58
**Mean Accuracy**	**71.66**	**73.44**	**77.87**	**79.94**	**85.05 ****

The numbers in bold represent the mean accuracy. ****** denotes the highest accuracy reported.

**Table 5 sensors-23-03474-t005:** The mean classification performance metrics with a 10-fold cross-validation technique across all participants in Experiment 1. Ten test runs were performed on each model, and the average of the results was calculated.

Classifier	Accuracy	Precision	Recall	F1-Measure	Specificity	NPV
LSTM	73.24	0.8187	0.7335	0.7651	0.7369	0.7249
CNN	76.23	0.8380	0.7657	0.7912	0.7818	0.7760
CNN-LSTM	78.95	0.8532	0.7862	0.8138	0.8172	0.8000
CNN-BiLSTM	81.49	0.8708	0.8135	0.8364	0.8353	0.8207
CNN-BiLSTM-CRF	85.65	0.8981	0.8601	0.8725	0.8688	0.8575

**Table 6 sensors-23-03474-t006:** The mean classification performance metrics with a 10-fold cross-validation technique across all participants in Experiment 2. Ten test runs were performed on each model, and the average of the results was calculated.

Classifier	Accuracy	Precision	Recall	F1-Measure	Specificity	NPV
LSTM	71.66	0.8075	0.7128	0.7514	0.7045	0.7387
CNN	73.44	0.8200	0.7328	0.7673	0.7434	0.7629
CNN-LSTM	77.87	0.8471	0.7832	0.8046	0.7558	0.7830
CNN-BiLSTM	79.94	0.8598	0.7948	0.8228	0.8449	0.8214
CNN-BiLSTM-CRF	85.05	0.8949	0.8434	0.8678	0.8570	0.8531

**Table 7 sensors-23-03474-t007:** Comparing our classification performance to previous EEG-based emotion recognition studies in financial trading.

Reference	Classifier	Accuracy (%)
[17]	Deep Learning and Random Forest	83.18
[10]	k-Nearest Neighbors(k-NN)	75.00
Our Proposed Method	CNN-BiLSTM-CRF	85.65

**Table 8 sensors-23-03474-t008:** Comparing the proposed framework with past studies.

Study	Model(s)	Feature Extraction	Participants	Advantages	Disadvantages
[17]	Multi-layer Feedforward ANN andRandom Forest	Differential Entropy (DE), Differential Asymmetry (DASM), and Rational Asymmetry (RASM).	10 (5 Males, 5 Females)	High accuracy (82.61%) with real market data and simulated money, feature correlation explained using statistical methods, and self-labeling performed with SAM.	Accuracy was reported as the only metric, with the performance of the deep learning algorithm lower than random forest. Little focus was on artifact removal.
[10]	kNN, Multi-layer Perceptron, and Random Forest	High-Order-Crossing (HOC)	8 (4 Males, 4 Females)	A simulated work environment for a competitive setting.	HVLA class was not considered, small dataset (eight participants), highest accuracy (75%) was lower compared to other relevant works.
[43]	Multi-layer Perceptron	Kernel Density Estimation (KDE)	1 Active investor	Displays emotion distribution in the valence-arousal quadrant and examines the impact of fear on investors’ rational decision-making.	Only one participant was considered and did not report any classification metrics obtained from the MLP model; only emotion distributions were reported.
Our work	LSTM, CNN, CNN-LSTM, CNN-BiLSTM, and CNN-BiLSTM-CRF	Independent Component Analysis (ICA)	20 (10 Males, 10 Females)	High accuracy (85.65% and 85.05%) in two experiments, robust to artifacts with multi-level removal, rigorous evaluation procedure, feature contribution explained with deepSHAP, and new findings on LVHA emotions and stop loss/limit orders.	Did not explore the applicability of the proposed architecture to other domains. The work used experienced traders and future research needs to expand the EEG dataset to a more diverse population including beginners.

## Data Availability

Data available upon request.

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
