# Peer review of "EEG-Based Emotion Classification in Financial Trading Using Deep Learning: Effects of Risk Control Measures"

_sensors, 2023, doi:10.3390/s23073474_

Round 1

Reviewer 1 Report

EEG is highly sensitive to the powerline, muscular, and cardiac artifacts. In EEG data preprocessing, authors need to mention how you handle AC power, ECG, and EMG artifacts in EEG signals. Same for EOG, EMG and others. Do the authors think their proposed method is robust to such artifacts?

Please write down the contribution of the study at the end part of the Introduction section in bulleted form.

Authors should improve the conceptual figures of their DL-proposed frameworks with more details and model parametrization.

What is the epoch length of EEG signal? Few figures are not easy to read  and quality need to improve.

Authors should introduce the EEG applications in ML-based disease, and mental workload prediction in broad scope, such as article, Explainable Artificial Intelligence Model for Stroke Prediction Using EEG Signal; in article, healthsos: real-time health monitoring system for stroke prognostics; in article, quantitative evaluation of task-induced neurological outcome after stroke; in article, driving-induced neurological biomarkers in an advanced driver-assistance system; and in article, quantitative evaluation of eeg-biomarkers for prediction of sleep stages.

The authors need to mention the model parameters or hyperparameters of DL models. The performance of the model is dependent on the selection of the architecture and/or parameters.

Authors should report more performance measures of classifiers, such as sensitivity, specificity, precision, and negative predictive value from the confusion matrix.

Both cross-validated training and testing ROC curves of all emotion classes.

I recommend to use deepSHAP/Grad-CAM to explain the contribution of EEG features in DL model.

The discussion section needs to be included. Authors must make discussion on the advantages and drawbacks of their proposed method with other studies adding a table in the discussion section.

Author Response

We have provided a point-by-point response to all review feedback and made the necessary changes as required by the Reviewers. Please refer to the attachment for details.

Reviewer 2 Report

The research paper discussed a very promising research area of Emotion Classification in Financial Trading. The following aspects of the study shall be focused and revised to enhance its quality level.

1. In the abstract, research statement and the research problem is not highlighted. The authors have simply narrated the story. It is recommended to explicitly mention/highlight the research problem which is focused in this study.
2. In objective 1, "To provide a technical framework for emotion classification in financial trading using 96 EEG data and deep learning algorithms",  framework for emotion classification in financial trading is proposed. The authors should explain how a generic emotion recognition/classification framework is different from the one used in financial trading. What aspects of the finance make the generic framework unusable.
3. Objective 2, "To identify the most appropriate neural network architecture for classifiers with gen- 98 eralization capabilities comparable to the state-of-the-art.", is about the selection of good NN architecture ...  What method is used for the selection of this NN/Deep learning model. If it is not done based on some meta characteristics of the data then how it can be claimed as a contribution. For guidance, the authors are recommended to study https://dl.acm.org/doi/pdf/10.1145/3164541.3164601 and use the approach, if applicable, should be adopted.
4. The sample/data size "data in this study to validate the self-re- 109 ported emotional states of twenty healthy individuals" used seems too small. If the authors could justify it with some valid references mentioning that this much data is enough in this kind of study, then it would be highly appreciated.
5. Section 2.3 Experiments, seems misplaced. It should be after the methodology section.

6. ???????? is used as the only performance metric which sometimes misleads the results. It is recommended that other measures, such as Precision, Recall, F-measure also be used. 

Author Response

(The authors gave the same response as above.)

Round 2

Reviewer 2 Report

All my comments are addressed. I am satisfied from the revision.

Author Response

The comments to Reviewer 2 had already been addressed in the previous round.